# An Italian Network of Population-Based Birth Cohorts to Evaluate Social and Environmental Risk Factors on Pregnancy Outcomes: The LEAP Study

**DOI:** 10.3390/ijerph17103614

**Published:** 2020-05-21

**Authors:** Teresa Spadea, Barbara Pacelli, Andrea Ranzi, Claudia Galassi, Raffaella Rusciani, Moreno Demaria, Nicola Caranci, Paola Michelozzi, Francesco Cerza, Marina Davoli, Francesco Forastiere, Giulia Cesaroni

**Affiliations:** 1Department of Epidemiology, ASL TO3 Piedmont Region, 10095 Grugliasco (TO), Italy; raffaella.rusciani@epi.piemonte.it; 2Regional Health and Social Care Agency, Emilia-Romagna Region, 40127 Bologna, Italy; nicola.caranci@regione.emilia-romagna.it; 3Centre for Environmental Health and Prevention, Regional Agency for Prevention, Environment and Energy of Emilia-Romagna, 40127 Bologna, Italy; aranzi@arpae.it; 4Unit of Clinical Epidemiology, Città della Salute e della Scienza University Hospital and Centre for Cancer Prevention (CPO), 10126 Turin, Italy; claudia.galassi@cpo.it; 5Regional Environmental Protection Agency, Piedmont, 10135 Turin, Italy; moreno.demaria@arpa.piemonte.it; 6Department of Epidemiology—Lazio Regional Health Service, ASL Roma 1, 00147 Roma, Italy; p.michelozzi@deplazio.it (P.M.); fracerza@gmail.com (F.C.); m.davoli@deplazio.it (M.D.); 7Institute for Biomedical Research and Innovation, National Research Council, 90146 Palermo, Italy; fran.forastiere@gmail.com; 8Science Policy & Epidemiology, King’s College, London SE1 9NH, UK

**Keywords:** socioeconomic status, air pollution, pregnancy outcomes, population-based birth cohorts

## Abstract

In Italy, few multicentre population-based studies on pregnancy outcomes are available. Therefore, we established a network of population-based birth cohorts in the cities of Turin, Reggio Emilia, Modena, Bologna, and Rome (northern and central Italy), to study the role of socioeconomic factors and air pollution exposure on term low birthweight, preterm births and the prevalence of small for gestational age. In this article, we will report the full methodology of the study and the first descriptive results. We linked 2007–2013 delivery certificates with municipal registry data and hospital records, and selected singleton livebirths from women who lived in the cities for the entire pregnancy, resulting in 211,853 births (63% from Rome, 21% from Turin and the remaining 16% from the three cities in Emilia-Romagna Region). We have observed that the association between socioeconomic characteristics and air pollution exposure varies by city and pollutant, suggesting a possible effect modification of both the city and the socioeconomic position on the impact of air pollution on pregnancy outcomes. This is the largest Italian population-based birth cohort, not distorted by selection mechanisms, which has also the advantage of being sustainable over time and easily transferable to other areas. Results from the ongoing multivariable analyses will provide more insight on the relative impact of different strands of risk factors and on their interaction, as well as on the modifying effect of the contextual characteristics. Useful recommendations for strategies to prevent adverse pregnancy outcomes may eventually derive from this study.

## 1. Introduction

The association between socioeconomic position and pregnancy outcomes is well established [1,2]. A variety of indicators of socioeconomic disadvantage, at both the neighbourhood and individual level, have been found significantly associated with adverse outcomes across countries and ethnic groups [3]. The most plausible causal mechanisms and mediating factors include smoking, mostly for intrauterine growth restriction, and genital infections and psychosocial factors, mainly for preterm births [1]. However, none of these behavioural determinants fully explain the observed social inequalities, and it is necessary to better understand their aetiology in pregnancy outcomes [4]. Moreover, a recent European comparative study highlighted educational inequalities in the risk of preterm birth in most of, but not all, the cohorts analysed [5], suggesting the presence of other potential determinants.

Another strand of research, in fact, focuses on contextual factors, meant as both socioeconomic characteristics of neighbourhoods and environmental pollution. As for the latter, the evidence produced so far on the association between air pollution and different birth outcomes is incomplete [6,7]. Recent findings add complexity to the picture, reporting evidence of an interactive effect of maternal behaviours and contextual factors on birthweight in Canada [8]. Systematic reviews pointed out that the heterogeneity or absence of association with air pollution reported in some studies may be due to difficulties in either quantifying exposure or adjusting for residential mobility [7,9].

Despite the amount of literature on the two key determinants, studies analysing them conjunctly in order to understand the single and combined impact of environmental and individual risk factors are scant [3,8]. Furthermore, most of the published studies rely on North American populations or on selected hospital-based birth cohorts, whose results are not directly generalizable to southern European population-based birth cohorts [10].

In a few Italian cities, population-based longitudinal cohorts are available. Starting from there, we established a network of population-based birth cohorts in the cities of Turin, Reggio Emilia, Modena, Bologna, and Rome (northern and central Italy), to study the role of air pollution exposure and socioeconomic status (SES) on term low birthweight, preterm births, small for gestational age, and pregnancy-induced hypertensive complications. In this first article, we outline the methodology and database of the study and the preliminary descriptive results.

## 2. Materials and Methods

### 2.1. Cohorts

The Italian network of population-based birth cohorts stems from the larger Italian Network of Longitudinal Metropolitan Studies (IN-LiMeS), a multicentre set of metropolitan population cohorts enrolled in nine Italian cities, namely Turin, Venice, Reggio Emilia, Modena, Bologna, Florence, Leghorn, Prato and Rome for monitoring socioeconomic inequalities in health [11]. The inclusion of Longitudinal Metropolitan Studies in the National Statistical Programme (NSP) complies with the national legislation on the processing of personal data for statistical and scientific research purposes, for each single longitudinal study (old and new releases of the NSP are available at https://www.sistan.it/index.php?id=52).

In order to identify birth cohorts and analyse the role of socioeconomic and environmental factors on pregnancy outcomes, we established a sub-network of cities with available data, resulting in 5 cities from northern and central Italy: Turin, three cities from the Emilia-Romagna (ER) region (Bologna, Reggio Emilia and Modena) and Rome. The Ministry of Health funded this project (grant RF-2011-02352442), named “Long term Exposure to Air pollution and Pregnancy outcomes”, the LEAP study.

The Italian network of population-based birth cohorts gathered demographic, socioeconomic and clinical information at individual level. The minimum core of population data to participate in the network included the following sources, linked to each other through an individual identification code: (1) birth certificates, (2) the municipal population register and (3) hospital discharge records. In addition, to participate in the LEAP study, we needed both geocoded residential addresses of women and small-scale models for residential air pollution exposures.

According to the availability of the data in all cities at the beginning of the project, we used the 2007–2013 birth cohorts and selected all singleton livebirths from women aged 15–49 years at delivery, who were residents in the 5 cities. From municipal registries, we retrieved information on the complete residential history of the women during the whole pregnancy from the date of the last menstruation, with the twofold objective of selecting only those who lived in the cities for the entire pregnancy and attributing them a more accurate level of air pollution exposure.

### 2.2. Socioeconomic, Demographic and Obstetric Characteristics

The dataset contained several maternal socioeconomic and demographic characteristics taken from the birth certificates, such as date of birth, level of education (university, high school, junior high school, primary school), occupational status (employed, unemployed, looking for the 1st occupation, student, housewife, other), marital status (not-married, married, separated, divorced, widowed), and citizenship (Italian, foreigner). It also included some information on maternal obstetric history and on pregnancy and new-borns: date of the last menstrual period, number of previous pregnancies/abortions, parity, gestational age, type of delivery, sex of the new-born, birthweight (g), length (cm), cranial circumference (cm) and Apgar score.

To account for contextual socioeconomic characteristics, we retrieved a composite indicator of deprivation at the census block level and linked it to the census block of the mother’s residence. This indicator, available for both 2001 and 2011 census data, included five elementary components, expression of both material and social deprivation, namely the standardized percentages of low education, unemployment, home tenancy, lone-parent family and overcrowding [12,13]. We then classified the continuous index in five categories, based on the percentiles of resident population within each city, so that each category represented about 20% of the city’s population, ordered by the deprivation score (1 = least deprived to 5 = most deprived).

As pregnancy outcomes to be investigated, we computed the following dummy variables: preterm births (births at <37 gestational weeks); low birthweight (i.e., <2500 g at birth) among term births; and small for gestational age (<10th percentile), based on the distribution of new-borns to Italian mothers, by infant sex, gestational age and parity [14]. Moreover, from the hospital discharge records we gathered information on maternal discharges during pregnancy, searching specifically for diagnoses of hypertension (ICD-9: 642), mild preeclampsia (ICD-9: 642.4), severe preeclampsia (ICD-9: 642.5), eclampsia (ICD-9: 642.6), or any preeclampsia-eclampsia (ICD-9: 642.4–642.7). The results on pregnancy-induced hypertensive complications, however, are outside the scope of this paper and will be fully reported in a subsequent paper.

### 2.3. Air Pollution Exposure Assessment 

#### 2.3.1. Standard Models

Data for Turin and Rome were derived from the European Study of Cohorts for Air Pollution Effects (ESCAPE), which provided Land use regression (LUR) models to estimate 2010 annual average concentrations of particulate matters (PM10, PM-coarse, PM2.5, PM2.5-absorbance), and nitrogen oxides (NO_2_ and NOx) [15,16]. Briefly, LUR models are useful to estimate the spatial variability of air pollution in urban areas, with the assumption that the spatial variability of pollutants’ concentrations does not change over time. In Turin and Rome, particulate matter of varying sizes was measured in 20 sites, and nitrogen oxides were measured in 40 sites in three separate two-week periods (to cover different seasons) over 2010. The three measurements were averaged to estimate the annual average of each pollutant, adjusting for temporal variation by using a centrally located background reference site, which was operating for a whole year. By using several variables (i.e., altitude, population density, industrial land use, green space, and traffic flow variables), city-specific land use regression (LUR) models were developed to explain the spatial variation of each measured pollutant. The R^2^ of the models ranged from 0.70 to 0.84 in Rome, and from 0.70 to 0.88 in Turin.

Using the same ESCAPE methodology, LUR models for nitrogen dioxide (NO_2_) have been developed also in Bologna, within the Strategic Programme Environment and Health (supported by the Italian Ministry of Health) and in Reggio Emilia and Modena within the LEAP study; the R^2^ of the models were 0.79, 0.72 and 0.78, respectively. As for PM10 and PM2.5, we used available European LUR models [17,18], together with dispersion models provided by the local Environmental Protection Agency to assign exposure values in Bologna, Modena and Reggio Emilia.

The city-specific models were then used to estimate the concentrations of air pollution at the residential coordinates of all woman’s addresses, and to calculate the individual average exposure weighted for the time of residence in each address.

#### 2.3.2. Back-Extrapolated Models

Since the time windows of vulnerability to environmental exposures could differ for each specific pregnancy outcome, and to take account of the slow and progressive decrease in air pollution concentrations in the last 15–20 years, we calculated back-extrapolated exposure during the entire pregnancy, in each trimester, in the first five months, and in the last week of pregnancy. In all the included cities, the concentrations of air pollutants (gaseous and particles) were slowly but progressively reducing over time during the last 15–20 years. LUR models have been developed for spatial variations under the hypothesis of their stability over time. Therefore, in order to account also for the observed temporal variations, we derived additional estimates of exposure with extrapolation techniques. Following literature suggestions [19], we used data from routine background monitoring stations (available for PM10, PM2.5 and NO_2_ from 2007) to temporally adjust the LUR estimates to the periods corresponding to each individual pregnancy and trimester of pregnancy. Briefly, after collection of daily air pollution data from background monitoring sites that cover the period 2007–2013 for each city, we followed these steps:We calculated the “annual” average concentration for the background monitoring sites during the measurement period used to build standard models (MLUR).For each day from 2007–2013, we calculated the ratio between the daily concentration (DC) and the annual average covering the LUR-models measurement period: Ratio = DC/MLUR.For each day, we calculated the extrapolated concentration by multiplying the modelled LUR concentration attributed to each subject (CLEAP) with the Ratio: C extrapolated = CLEAP × Ratio.We calculated both an average exposure during pregnancy, using extrapolated temporally adjusted exposures, and trimester specific exposures.In order to account for the nine months of pregnancy, we attributed these back-extrapolated exposures only to births occurred from 2008 to 2013.

For brevity reasons, however, unless otherwise specified, in this paper we will refer to the estimates derived from the standard models.

#### 2.3.3. Additional Exposure Models

In addition to particulate matter and nitrogen oxides, in Turin and Rome, LUR models for metal components of PM10 and PM2.5 were available from the TRANSPHORM project [20]. The elements were selected based upon evidence for health effects (toxicity), a high percentage of detected samples (>75%), and representation of major anthropogenic sources. We selected Cu, Fe, and Zn mainly for (non-tailpipe) traffic emissions; Ni and V for mixed oil burning/industry; S for long-range transport; Si for crustal material; and K for biomass burning. The elements chosen do not necessarily represent single sources. The measurements were taken at the same time of particulate matter, and the LUR models have been developed centrally. The models performed better for PM10 than for PM2.5 components: the average R^2^ for elements in PM10 and PM2.5 were in Rome 0.75 and 0.75 respectively, and in Turin 0.76 and 0.63, respectively [20].

Estimates of exposure to particulate matter constituents in Turin and Rome are included in the full database, but they will not be further illustrated here, as we will focus only on data available in all cities.

## 3. Results

### 3.1. Cohorts’ Selection and Description

During the study period, the five cities included in the study registered 227,497 birth certificates related to singleton livebirths from women aged 15–49 years at delivery, who were resident in the same city for the entire pregnancy (Table 1). Coherently with the population size in each city, more than 60% of the entire cohort came from Rome, 20% from Turin and less than 20% from the three cities of Emilia-Romagna region. Individual record-linkage procedures for geocoding and exposure assessment had very high success rates, all above 90% in all cities. After a few exclusions for either the presence of malformations or missing information in outcome variables, we were left with 211,853 birth certificates to be analysed, corresponding to 93.1% of the total birth certificates initially retrieved. The number of births remained constant in the study period, apart from a slight decline in births in the last year in all cities except Bologna.

Table 2 illustrates all the individual information gathered on the mothers and the births, which can be used as potential determinants of the pregnancy outcomes, or confounders in the association between air pollution and pregnancy outcomes. There were a few missing data for sociodemographic characteristics, but the percentages for each variable in each city were always less than 2%, except for maternal education in Turin, which was unknown in 5% of cases (data are not displayed in the table but their numbers can be derived as a difference from the total). The mean maternal age was around 32–33 years in all cities. Overall, women were mainly highly educated (27% with a university degree and 47% with a high school diploma) and employed (71%); two thirds were married and 20% were foreigners. Slightly more than half (54%) were at their first delivery and more than one third (38%) had a caesarean section. These characteristics were not uniformly distributed across cities: Bologna had the highest percentages of graduated, employed and not married women (47%, 81% and 35%, respectively), Turin had the highest percentage of immigrants (31%), while in Rome women with primary education (8%), unemployed (10%) and those who underwent a caesarean section (43%) were overrepresented compared to the average.

### 3.2. Air Pollution Exposure

Table 3 illustrates the exposure of the women under study during pregnancy in the five cities, according to the standard models. Turin had the highest level of exposure for PM10, PM2.5 and NO_2_. Modena had the highest level of PM2.5–10 (17.1 µg/m^3^ average exposure), and Rome had higher levels of NO_2_, a traffic-related pollutant, compared to cities in Emilia-Romagna.

Using the back-extrapolated concentrations of PM10, PM2.5 and NO_2_ (data not shown), we observed a high variability of exposure during pregnancy according to the season of conception, reflecting the seasonal trends of nitrogen oxide and particle concentrations, with highest levels during winter, lowest in summer and intermediate in spring/autumn. Women who started their pregnancy in winter had the highest levels of exposure during the first trimester of pregnancy and the lowest during the second trimester. Those who started the pregnancy in spring had levels of exposure that increased from the first to the third trimester. Women who started the pregnancy during summer had low levels of exposure in the first trimester, and high levels in the second and third trimester. In the case of autumn conceptions, women had decreasing levels of exposure from the first to the third trimester of pregnancy.

### 3.3. Outcomes

The main pregnancy outcomes are illustrated in Table 4. The mean values of the continuous variables were quite homogeneous across cities, with a mean gestational age of 39 weeks, birthweight of around 3.3 kilos, length and cranial circumference of 50 and 34 centimetres, respectively. However, we can appreciate a few differences looking at the categorical outcomes, derived as above or below specific thresholds (as specified in the methods). In particular, in Turin, we observe a higher percentage of term low birthweight and small for gestational age new-borns (2.5% and 9.5% compared with an average of 2.2% and 8.6%, respectively). Reggio Emilia and Modena have the highest percentages of large for gestational age infants, and finally, in Modena we can observe a slight excess of new-borns who need assistance or resuscitation (1% vs. 0.5% overall).

### 3.4. Interactions between Determinants

As a first exploratory analysis, we looked at the possible interaction between the two strands of determinants of pregnancy outcomes, i.e., socioeconomic status (SES) and air pollution. This descriptive exploration is useful as a background to inform future work, in which we will analyse their impact on pregnancy outcomes. For the sake of simplicity, we selected only three indicators of air-pollution exposure, namely PM10, PM2.5 and NO_2_ for the entire pregnancy and four SES indicators: maternal education, maternal occupation (contrasting employed vs. not employed, i.e., unemployed, looking for the first job and housekeepers), citizenship and the 2011 census block deprivation index. In the following graphs (Figure 1, Figure 2, Figure 3 and Figure 4), we show the prevalence of high air pollution exposure (>75th percentile of the distribution) by each SES indicator and by city. The results appear highly heterogeneous across pollutants and across cities.

Looking separately at each city, in Turin, we observe a significant inverse correlation with all indicators of SES (i.e., highest levels of pollution among lower social classes) only for PM10, although in the lowest category of both education and deprivation, there is the tendency to a decline in the levels of PM10 compared to the previous class. On the contrary, NO_2_ shows a definite direct association with SES (higher levels among more affluent classes) and PM2.5 has a pattern similar to NO_2_, but with less steep gradients.

Bologna displays a more consistent pattern, with a clear inverse association of all the SES indicators with both PM10 and PM2.5, and an equally clear direct gradient for NO_2_.

Analogously, the association between pollution and SES is particularly evident in Reggio Emilia: all the SES indicators have an inverse gradient for all the pollutants, with the only exception of the U-shaped curve of PM10 and NO_2_ for maternal education (with a higher exposure among women with a university degree than among those with a high school diploma) (Figure 1).

In Modena, on the contrary, the patterns are variable across pollutants and SES indicators. Specifically, PM10 and NO_2_ have an inverse association with maternal occupation (Figure 2) and citizenship (Figure 3), a reversed U-shaped curve for deprivation (higher levels of exposure in the middle classes) (Figure 4), and a direct U-shaped curve for education (Figure 1); on the other side, PM2.5 has a reversed U-shaped curve for education and deprivation, and non-significant differences by occupation and citizenship.

Finally, in Rome, there is consistency in the patterns of the three pollutants, but different directions of the association across SES indicators: a U-shaped curve for maternal education, similar levels of pollution exposures by occupation, higher levels for foreigners (inverse association), and decreasing pollution with increasing deprivation, with a tendency to increase again among the most deprived women.

## 4. Discussion

We ended up with a pooled cohort including more than 200,000 births, in five Italian cities and seven years of observation. It is the largest Italian population-based birth cohort, built on individual record-linkage between current administrative databases, censuses and health information systems, which has the advantage of being economic and sustainable over time. The very high success rates at linkage guarantee that the cohorts include all births in the area of interest and are not distorted by selection mechanisms or social desirability bias, such as birth cohorts based on voluntary enrolment and/or face-to-face interviews on socioeconomic characteristics. Looking forward, also the follow-up of mothers and new-borns in the first years of life will be possible, through linkage with hospitalizations, pharmaceutical prescriptions and outpatient services. Moreover, this cohort is placed in a setting characterised by higher levels of air pollution, compared to North America or other European countries, and therefore, it will provide estimates of the impact on pregnancy outcomes of less investigated levels of exposure. In addition, the back extrapolated models will provide time window-specific estimates to account for the outcome-specific windows of vulnerability to environmental exposure during pregnancy.

We have observed little differences in both the determinants and the outcomes across cities, but they can be appropriately taken into account in the multivariable analysis. More interestingly, we have observed that the association between socioeconomic characteristics and air pollution exposure do not go in the same direction everywhere. This is in line with previous literature, underlining that often in big cities (as Rome), people with a high SES are highly exposed to air pollution, since they tend to live in the inner and most polluted areas of the city [21]. On the contrary, in our smaller cities (such as Turin and Bologna), in addition to the rich central districts, there are many affluent neighbourhoods in the greenest hilly areas. Furthermore, exposure to vehicular traffic can also vary depending on the presence or absence of pedestrian areas in the centre of the cities included in the study. Therefore, results can be correctly interpreted only after a thorough analysis of the orographic-urbanistic characteristics of each city.

From our initial analysis however, we can draw a first picture that can explain some of the observed differences. The cities in the North (Turin, Modena, Reggio Emilia and Bologna) are located in the Po Valley, the most polluted area of Italy. For its orography, high air pollution concentrations, produced by several industries, in addition to heating and vehicular traffic, tend to stagnate in the valley. On the contrary, Rome is not an industrial city, it is not surrounded by mountains, and has a mild temperature, with breezes from the Tyrrhenian Sea. Given its characteristics, air pollution in Rome is mainly due to inner vehicular traffic. The differences we found in the distribution of high levels of exposure by SES between the two macro-areas might reflect the main sources of air pollution: in fact, in Rome, the source is mainly the same for all pollutants (traffic), while in the northern cities, PM10 can represent heating or industrial emissions, NO_2_ vehicular traffic, and PM2.5 a mixture of the two. The LUR methodology provides equations that depend on the measured concentrations and the city specificities. Therefore, in each city, the variables which enter the models can differ even in equations of similar pollutants, such as particulate matter (PM10, PM2.5, and PM2.5–10). In Turin, for example, PM2.5 is affected among the others by the traffic load on major roads, producing high concentrations along inner avenues, where high-class people live, whereas the PM10 equation includes also the population density, thus reducing the estimated concentrations in the central area of the city and accentuating its levels in the more deprived western area.

The geographical differences between the cities in the Po Valley and Rome suggest the need to perform city-specific analyses, and, in a comparative approach, to consider the city as an effect modifier of the association between air pollution exposure and health outcomes. Moreover, the diverse patterns of exposure by the different SES indicators also suggest a possible effect modification of the socioeconomic position on the impact of air pollution on pregnancy outcomes, which must be carefully considered in the future analyses. In summary, there are only two cases of consistency for the different SES indicators. First, we observed that for almost everywhere, the levels of all pollutants were higher among non-Italian women, suggesting a specific area of prevention intervention that might be soon undertaken. Secondly, maternal occupation had the most similar values of pollution in the two classes considered, and this might indicate the necessity of considering a more detailed classification of occupation, which was not possible in this study for the smaller cities.

The differences in exposure by season of conception are higher than spatial differences within cities (data not shown), thus suggesting that spatial variability of concentrations might be less important than temporal variability in city-specific models. As it has been suggested [22], an accurate choice of time-windows of exposure is necessary for each single outcome, depending on the specific pollutant and on the hypothesised mechanisms of association. We will therefore perform sensitivity analyses to identify the best exposure assessment in studying the association between air pollution and pregnancy outcomes.

Our study has some limitations. Firstly, the birth cohort represents only three regions in the North and Centre of the country; being based on current information systems, however, the model is easily reproducible and transferable to other areas. Compared to hospital-based birth cohorts, we lack data from medical records and on lifestyles potentially related to the outcomes under examination, such as smoking, type of job or body mass index. The socioeconomic characteristics, however, being in turn associated with lifestyles, allow us to analyse the association between pollution and pregnancy outcomes, with a partial adjustment also for these risk factors. Moreover, although the success rate at linkage is overall high, as well as the completeness of the datasets, there is some variability across cities in the percentages of unlinked records and missing data. This needs to be further investigated in order to understand the mechanisms involved and avoid possible biases [23].

There are also limitations regarding the exposure assessment. The approaches are completely comparable among cities only for NO_2_, for which we have adopted exactly the same kind of model with the same protocol. Different approaches, with different degrees of reliability, have been applied for PMs in the cities of Emilia-Romagna, compared to Turin and Rome. Moreover, using the standard LUR models providing estimates for each maternal address, we assume that, besides the decreasing levels of air pollution concentrations in recent decades, the spatial contrasts of pollutants do not vary over time within the cities. This was the case for NO_2_ in Rome [24] and in other settings [25,26], but we do not have evidence for particulate matter, nor for the cities of the Po Valley. On the other side, applying the extrapolation techniques, the high values in the standard deviations of the extrapolated data suggest a high degree of uncertainty of the estimates. This procedure is at present one of the most innovative methodologies, agreed upon at the European level; it is aimed at defining exposure gradients that make data independent of external time trends and comparable over the entire observation period. Problems of potential overestimation (or underestimation) of the exposure related to the procedure are possible, but we standardized the methodology across cities in order to make this possible bias non-differential, therefore not affecting the point estimate of the risks.

## 5. Conclusions

Adverse birth outcomes are the major cause of infant morbidity and mortality. At present, the literature on the association between air pollution exposure and pregnancy outcomes and on the aetiology of social inequalities in pregnancy outcomes is still inconclusive. Our study will be the first Italian study to evaluate the impact of air pollution on pregnancy outcomes in a long-term approach, evaluating also the role of socioeconomic factors. Results from the ongoing multivariable analyses will provide more insight on the relative impact of different strands of risk factors and on their interaction, as well as on the modifying effect of contextual characteristics. Moreover, the possibility of disentangling the effect of air pollution exposure according to the specific season of conception and trimester of pregnancy will provide further data for in-depth research questions and possibly useful recommendations for strategies aimed at preventing adverse pregnancy outcomes.

## Figures and Tables

**Figure 1 ijerph-17-03614-f001:**
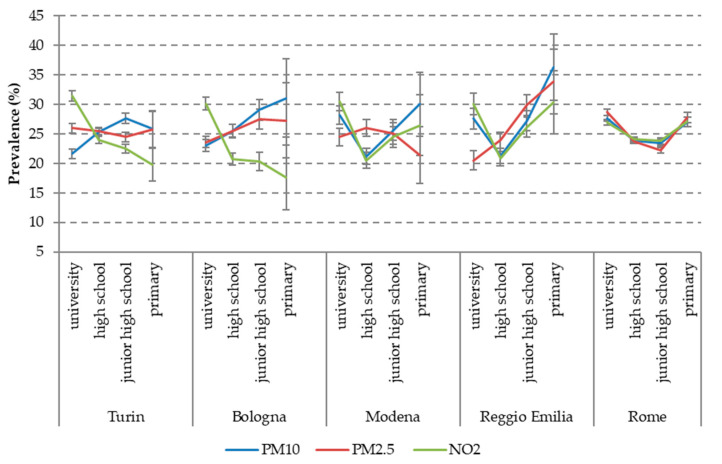
Prevalence of high air pollution exposure (>75th percentile) by city and maternal education.

**Figure 2 ijerph-17-03614-f002:**
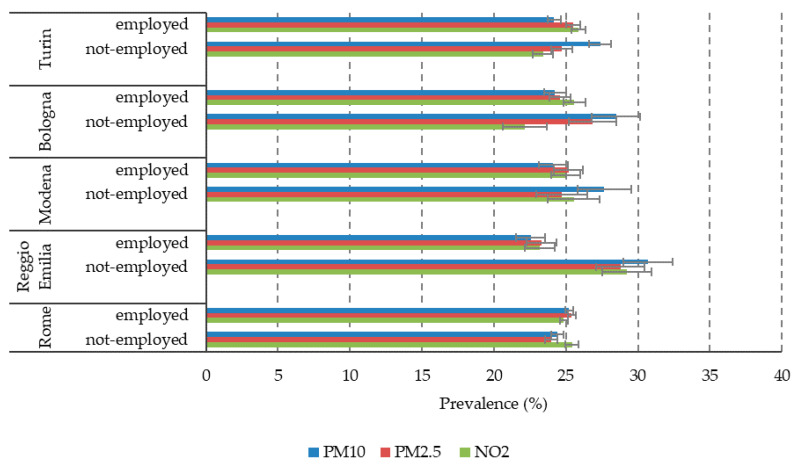
Prevalence of high air pollution exposure (>75th percentile) by city and maternal occupation.

**Figure 3 ijerph-17-03614-f003:**
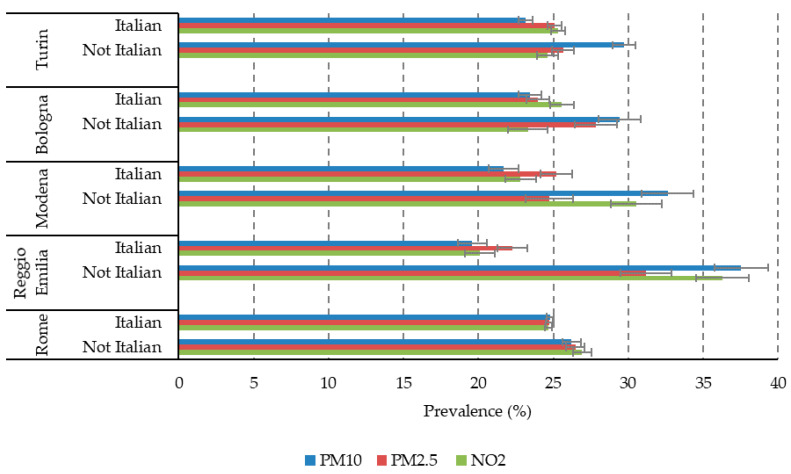
Prevalence of high air pollution exposure (>75th percentile) by city and maternal citizenship.

**Figure 4 ijerph-17-03614-f004:**
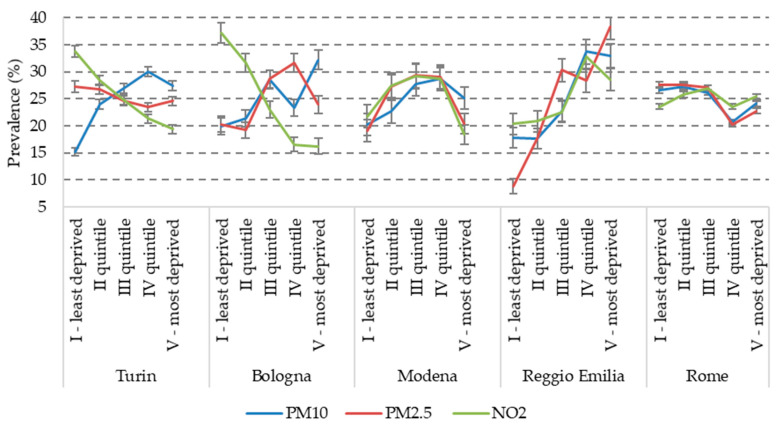
Prevalence of high air pollution exposure (>75th percentile) by city and 2011 census block deprivation index.

**Table 1 ijerph-17-03614-t001:** Selection of the birth cohorts 2007–2013.

Data Selection	Turin	Bologna	Modena	Reggio Emilia	Rome	Total
**N singleton livebirths**	46,376	16,817	9502	10,188	144,614	227,497
**Exclusions**						
missing geocoding	502 (1.1%)	524 (.1%)	16 (0.2%)	704 (6.9%)	6182 (4.3%)	7928 (3.5%)
missing exposure	195 (0.4%)	157 (0.9%)	14 (0.1%)	140 (1.4%)	3724 (2.6%)	4230 (1.9%)
*% success at linkage*	98.5	96.0	99.7	91.7	93.1	94.6
malformations at birth	803 (1.7%)	145 (0.9%)	46 (0.5%)	90 (0.9%)	2251 (1.6%)	3335 (1.5%)
missing outcomes *	32 (0.1%)	110 (0.7%)	5 (0.1%)	2 (0.0%)	2 (0.0%)	151 (0.1%)
**N births analysed**	44,844	15,881	9421	9252	132,455	211,853
*% of initial dataset*	96.7	94.4	99.1	90.8	91.6	93.1
**Year of delivery**						
2007	6421	2129	1364	1341	19,124	30,379
2008	6626	2127	1340	1280	18,590	29,963
2009	6649	2255	1359	1384	18,457	30,104
2010	6449	2288	1326	1364	19,978	31,405
2011	6429	2350	1367	1373	19,383	30,902
2012	6313	2349	1390	1270	19,011	30,333
2013	5957	2383	1275	1240	17,912	28,767

* birthweight, gestational age, infant sex, parity.

**Table 2 ijerph-17-03614-t002:** Maternal and birth characteristics.

Characteristics	Turin		Bologna		Modena		Reggio Emilia		Rome		Total	
	(*N* = 44,844)		(*N* = 15,881)		(*N* = 9421)		(*N* = 9252)		(*N* = 132,455)		(*N* = 211,853)	
**Maternal age, years** **m (SD)**	32.3 (5.3)		33.2 (5.2)		32.5 (5.3)		31.6 (5.4)		33.3 (5.2)		33.0 (5.2)	
	*N*	%	*N*	%	*N*	%	*N*	%	*N*	%	*N*	%
**Maternal education**												
university	9782	23.0	7404	46.6	3324	35.3	2422	26.2	34,133	25.8	57,065	27.3
high school	20,674	48.6	5741	36.1	3564	37.8	3976	43.0	63,501	48.0	97,456	46.5
junior high school	11,296	26.6	2549	16.1	2253	23.9	2570	27.8	24,188	18.3	42,856	20.5
primary	752	1.8	187	1.2	280	3.0	284	3.0	10,445	7.9	11,948	5.7
**Maternal occupational status**												
employed	30,501	69.1	12,775	80.7	7021	74.8	6326	68.4	92,764	70.1	149,387	70.8
unemployed	3471	7.9	497	3.1	310	3.3	429	4.6	12,884	9.7	17,591	8.3
looking for the 1st job	413	0.9	5	0.0	6	0.1	10	0.1	1393	1.1	1827	0.9
student	390	0.9	205	1.3	123	1.3	163	1.8	2124	1.6	3005	1.4
housewife	9163	20.7	2329	14.7	1919	20.4	2310	25.0	22,590	17.1	38,311	18.2
other	222	0.5	12	0.1	7	0.1	11	0.1	533	0.4	785	0.4
**Marital status**												
married	30,077	68.1	9913	63.4	6737	71.6	6589	71.4	86,773	65.5	140,089	66.4
not married	12,111	27.4	5534	35.4	2441	25.9	2448	26.5	42,194	31.9	64,728	30.7
separated	1185	2.7	90	0.6	135	1.4	107	1.1	2144	1.6	3661	1.7
divorced	689	1.6	69	0.4	88	0.9	81	0.9	1194	0.9	2121	1.0
widowed	70	0.2	21	0.1	9	0.1	6	0.1	149	0.1	255	0.1
**Citizenship**												
Italian	31,001	69.1	11,844	74.6	6603	70.1	6436	69.6	112,923	85.3	168,807	79.7
foreigner	13,843	30.9	4037	25.4	2818	29.9	2816	30.4	19,529	14.7	43,043	20.3
**Parity**												
first child	23,648	52.7	8788	55.3	4516	47.9	4307	46.6	74,063	55.9	115,322	54.4
second child	16,234	36.2	5517	34.7	3501	37.2	3674	39.7	45,979	34.7	74,905	35.4
>2nd child	4962	11.1	1576	9.9	1404	14.9	1271	13.7	12,413	9.4	21,626	10.2
**Type of delivery**												
natural unassisted childbirth	29,178	65.1	10,484	66.0	6551	69.5	6317	68.3	70,897	53.5	123,427	58.3
caesarean section	13,823	30.8	4708	29.6	2628	27.9	2559	27.7	56,976	43.0	80,694	38.1
assisted childbirth	1829	4.1	689	4.3	242	2.6	376	4.1	4458	3.4	7594	3.6
other	14	0.0	0	0.0	0	0.0	0	0.0	123	0.1	137	0.1
**Infant sex**												
boys	23,021	51.3	8214	51.7	4880	51.8	4764	51.5	68,207	51.5	109,086	51.5
girls	21,823	48.7	7667	48.3	4541	48.2	4488	48.5	64,248	48.5	102,767	48.5

**Table 3 ijerph-17-03614-t003:** Estimated exposure during pregnancy (Mean (SD)).

Pollutant	Turin	Bologna	Modena	Reggio Emilia	Rome
PM10 (µg/m^3^)	46.7 (4.5)	35.8 (2.3)	38.7 (2.6)	37.2 (3.0)	36.3 (4.9)
PM2.5–10 (µg/m^3^)	16.6 (3.0)	16.4 (2.0)	17.1 (2.3)	15.6 (2.5)	16.5 (3.3)
PM2.5 (µg/m^3^)	26.3 (1.3)	19.3 (0.7)	21.6 (0.6)	21.7 (0.8)	19.4 (1.8)
NO_2_ (µg/m^3^)	52.3 (9.1)	39.7 (6.4)	39.7 (7.3)	33.9 (9.1)	40.7 (10.7)

**Table 4 ijerph-17-03614-t004:** Pregnancy outcomes.

Outcomes	Turin			Bologna			Modena			Reggio Emilia			Rome			Total		
	(*N* = 44,844)			(*N* = 15,881)			(*N* = 9421)			(*N* = 9252)			(*N* = 132,455)			(*N* = 211,853)		
**Gestational age, weeks**																		
m (SD)	39.0 (1.8)			39.0 (1.9)			39.0 (1.8)			39.0 (1.8)			38.9 (1.7)			38.9 (1.8)		
**Birthweight in grams**																		
m (SD)	3256 (496)			3288 (510)			3307 (517)			3322 (523)			3252 (485)			3261 (493)		
**Length in cm**																		
m (SD)	49.4 (2.6)			50.0 (2.6)			50.1 (2.5)			50.3 (2.8)			50.0 (2.4)			49.9 (2.5)		
**Cranial circumference in cm**																		
m (SD)	34.0 (2.9)			34.3 (1.8)			34.2 (1.6)			34.4 (2.3)			34.3 (1.7)			34.2 (2.1)		
Outcomes	***N***	**%**	**(95% CI)**	***N***	**%**	**(95% CI)**	***N***	**%**	**(95% CI)**	***N***	**%**	**(95% CI)**	***N***	**%**	**(95% CI)**	***N***	**%**	**(95% CI)**
**Preterm birth**																		
no	41,994	93.6	(93.4–93.9)	14,879	93.7	(93.3–94.1)	8804	93.5	(92.9–94.0)	8655	93.5	(93.0–94.1)	124,834	94.2	(94.1–94.4)	199,166	94.0	(93.9–94.1)
yes	2850	6.4	(5.5-7.3)	1002	6.3	(4.8-7.8)	617	6.5	(4.6-8.5)	597	6.5	(4.5-8.4)	7621	5.8	(5.2-6.3)	12,687	6.0	(5.9–6.4)
**On term low birthweight**																		
no	40,961	97.5	(97.4–97.7)	14,595	98.1	(97.9–98.3)	8643	98.2	(97.9–98.5)	8469	97.9	(97.5–98.2)	122,067	97.8	(97.7–97.9)	194,705	97.8	(97.7–97.8)
yes	1033	2.5	(1.5-3.4)	284	1.9	(0.3-3.5)	161	1.8	(0.0-3.9)	186	2.1	(0.1-4.2)	2767	2.2	(1.7–2.8)	4431	2.2	(1.8–2.7)
**Weight for gestational age**																		
appropriate	36,183	80.7	(80.3–81.1)	12,862	81.0	(80.3–81.7)	7614	80.8	(79.9–81.7)	7429	80.3	(79.4–81.2)	108,540	81.9	(81.7–82.2)	172,628	81.5	(81.3–81.7)
small	4257	9.5	(8.6–10.4)	1237	7.8	(6.3–9.3)	698	7.4	(5.5-9.4)	680	7.3	(5.4-9.3)	11,246	8.5	(8.0–9.0)	18,118	8.6	(8.1–9.0)
large	4404	9.8	(8.9–10.7)	1782	11.2	(9.8–12.7)	1109	11.8	(9.9-13.7)	1143	12.4	(10.4-14.3)	12,669	9.6	(9.1–10.1)	21,107	10.0	(9.6–10.4)
**Apgar score**																		
7–10 (normal)	43,663	99.3	(99.3–99.4)	15,766	99.5	(99.4–99.6)	9330	99.0	(98.9–99.3)	9204	99.5	(99.3–99.6)	129,724	99.6	(99.6–99.6)	207,687	99.5	(99.5–99.5)
4–6 (needs assistance)	237	0.5	(0.0-1.5)	62	0.4	(0.0–1.9)	81	0.9	(0.0–2.9)	45	0.5	(0.0–2.5)	348	0.3	(0.0–0.8)	773	0.4	(0.0–0.8)
1–3 (needs resuscitation)	56	0.1	(0.0–1.1)	14	0.1	(0.0–1.6)	8	0.1	(0.0–2.1)	3	0.0	(0.0–2.1)	167	0.1	(0.0–0.7)	248	0.1	(0.0–0.5)

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
