# Peer review of "An Italian Network of Population-Based Birth Cohorts to Evaluate Social and Environmental Risk Factors on Pregnancy Outcomes: The LEAP Study"

_ijerph, 2020, doi:10.3390/ijerph17103614_

Round 1

Reviewer 1 Report

Thanks to the authors for the efforts. It is not an easy task as the study design is quite complex.

One of my major concerns is the selection of birth cohorts. If the data for air pollution monitoring stations was only available from year 2008, including birth cohort of year 2007 in the study appears irrelevant as the effects from air pollution during the pregnancy period (from year 2006-2007) was not available. 

My second concern is section 3.4. I understand the authors tried to look at the possible interactions between SES and air pollutions in different parts of Italy. However the reasons to work out these interactions were not clearly explained. Further ado, how do these interactions affect or correlate with the pregnancy outcomes in different regions were not analyzed and the readers may feel part of the analysis is missing before going on to the Discussion part.

Lastly in the Discussion part, the authors compared and contrasted the Po Valley (4 regions) versus Rome. Although the air pollutions characteristics seemed to differ in this way, the pregnancy outcomes among the 4 regions were actually varied greatly. For example, Reggio Emilia had the most number of LGA babies while Turin had the most SGA babies which the authors did not address. It is difficult for the readers to understand how the authors can make a conclusion that air pollution has impact on pregnancy outcomes, while the analyses only showed diverse findings.

Minor typo error at line 37 "1maternal".

It may be clearer to use "trimester" instead of "term" at line 43-44.

Reviewer 2 Report

This manuscript describes a population-based cohort focused on five Italian cities, emphasizing data on pregnancy, childbirth, sociodemographics, and air pollution exposures during pregnancy.  The text is generally well-written (a few comments below about items for correction), and the manuscript flows fairly well.  There are some areas where the manuscript could be improved, as the authors do not take a broad perspective on how their data repository adds to the considerable knowledge base of research internationally on the topic of interest.  

To start, while this might be the largest cohort of its kind in Italy, it most certainly is not so in the international context.  There are numerous studies of birth outcomes and air pollutants in North America, western Europe, Australia and elsewhere, many of which comprise much larger cohorts than that described here.  What is it specifically that the authors hope to better understand from this data application, given previous research?  What overall aims guided its creation?  These should be outlined in setting up this discussion of the cohort itself.

More detail is also needed concerning how components of the database are integrated by geography.  Yes, there are geocodes, and it is mentioned that sociodemographics were measured at the census block level.  But what addresses are used - does the register have access to the mother's residential history from preconception to delivery or is only a single address at birth?  And if so, how was this information used to develop more sensitive measures of exposure?

Similarly, more detail on how exposure measurements were made, and how estimates of specific maternal exposures were calculated would be helpful.  In preparing a similar study, my research team identified seven different decisions researchers must make in calculating estimates of maternal exposure, and published a paper showing how much difference these decisions can make (Tanner JP et al, Spat Spatio-Temp Epidemiol 17 (May 2016), 117-129.  PMID: 27246278).  

Given the relative sizes of each city cohort, it would be advisable to control for site in any overall analysis, with Rome seriously overweighting any study while three of the other cities are quite small.  

It would also be advisable to assess the effects of unlinked records and missing data.  Given that ungeocodable records cannot contribute directly to air pollution study or analyses involving broader sociodemographics, it would be important to understand what potential biases result.  Consider Doidge JC, Harron KL, Int J Epidemiol 48 (2019) 2050-2060.

The authors also do not discuss their methodological approach to contextual analysis, but census blocks are not statistically independent units, and some approach to spatially-enabled multi-level modelling will be necessary to fully assess any observed associations between birth outcomes, maternal and sociodemographic factors, and pollution exposures.

More specific comments follow:

p 1 line 39 - multivariable is more appropriate than multivariate.  Nowadays, epidemiologists use the former what there are many covariates or independent variables, and the latter only when multiple dependent variables are analyzed simultaneously.  Since analyses of this nature are not presented, the reviewer presumes the researchers should use the word 'multivariable' throughout.

p 2 line 6 - have instead of has

line 10 use active voice instead of 'there is a need to better understand'

line 28 established instead of set-up, also elsewhere

line 31-32 instead say 'in this article we outline . . . and preliminary descriptive results

p 3 line 24  nice to have head circumference and crown-heel length, but how standardized were the measurements?  Is there an established protocol followed in all hospitals?

p 4 as mentioned above, not clear what choices were made in determining how to interpolate data from monitoring stations to maternal address location, time windows for exposure, and other decisions

p 5 Table 1 are there functioning birth defects registries for each of these cities - it would be better to link to those data rather than taking congenital malformations from the birth hospitalization.  Typically the prevalence of major birth defects is about 3-5% of live births, much higher than what is reported here.

p 6 Table 2 citizenship is fine, but better would be nation of birth (or nativity).  Nativity has been associated with numerous adverse pregnancy outcomes.

p 9 so why do all of the cities have less SGA than the standard (10%)?

p 10 this section requires elaboration.  This clearly requires some form of multi-level modeling.  What software, what method, what decisions about intercepts, random vs fixed effects, etc?

p 11-12 Figures 1 and 3 need Y-axis label.  Can any generalizations be made based on these graphics?  What sort of multivariable analysis would help for that?

This reviewer did not carefully review the discussion section, as it will likely look different if suggestions made above are carried out.  Key would be to encapsulate the observations about the Italian data into the broader context of the international literature.

In the references, several are in capital case, and need to be corrected.  This includes ref 14 and 19.  But generally, this list of references misses most of the significant literature related to the general topic of this paper.

Reviewer 3 Report

Thank you for the opportunity to review the manuscript “An Italian network of population-based birth cohorts to evaluate social and environmental risk factors on pregnancy outcomes: the LEAP study” for International Journal of Environmental Research and Public Health. This is a thought-provoking manuscript, and I enjoyed reviewing it. The data are unique and allow the author(s) to examine the relationship between socioeconomic factors and air pollution exposure on several pregnancy outcomes. Importantly, in what I view as a major contribution to the literature, the authors use Italian population-based data to address their questions. Overall, it was clearly written and thought provoking. The literature was adequately reviewed; the data and methods were clearly articulated; the findings are well explained and supported by the models; the discussion is well grounded and situates the manuscript into the larger discussion. I have only praise for the manuscript.
